A systematic review of methods for studying consumer health YouTube videos, with implications for systematic reviews

Sampson Margaret 1 msampson@cheo.on.ca
Cumber Jordi 2
Li Claudia 3
Pound Catherine M. 4 5
Fuller Ann 6
Harrison Denise 2 7 8 9
1 Library Services, Children’s Hospital of Eastern Ontario , Canada
2 Children’s Hospital of Eastern Ontario Research Institute , Canada
3 Odette Cancer Centre, Sunnybrook Health Sciences Centre , Canada
4 Department of Pediatrics, Faculty of Medicine, University of Ottawa , Canada
5 Department of Pediatrics, Division of Consulting Pediatrics, Children’s Hospital of Eastern Ontario , Canada
6 Public Relations, Children’s Hospital of Eastern Ontario , Canada
7 School of Nursing, Faculty of Health Sciences, University of Ottawa , Canada
8 Murdoch Childrens Research Institute , Australia
9 Melbourne School of Health Sciences, University of Melbourne , Australia
Muench Fred
Electronic publication date: 2013 Sep 12
Publication date: 2013
Volume: 1
Electronic Location ID: e147
Received 2012 Dec 6; Accepted 2013 Aug 6
Copyright: © 2013 Sampson et al.
Copyright year: 2013
Copyright holder: Sampson et al.
License: This is an open access article distributed under the terms of the Creative Commons Attribution License, which permits unrestricted use, distribution, and reproduction in any medium, provided the original author and source are credited.
License URL: https://creativecommons.org/licenses/by/3.0/

Keywords: Social media, YouTube, Consumer health, Research methods, Systematic review

Funding: Summer Studentship grant from the Children’s Hospital of Eastern Ontario Research Institute The research was supported through a Summer Studentship grant from the Children’s Hospital of Eastern Ontario Research Institute. The funders had no role in study design, data collection and analysis, decision to publish, or preparation of the manuscript.

==============================
Background. YouTube is an increasingly important medium for consumer health information – with content provided by healthcare professionals, government and non-government organizations, industry, and consumers themselves. It is a rapidly developing area of study for healthcare researchers. We examine the methods used in reviews of YouTube consumer health videos to identify trends and best practices.

Methods and Materials. Published reviews of consumer-oriented health-related YouTube videos were identified through PubMed. Data extracted from these studies included type of journal, topic, characteristics of the search, methods of review including number of reviewers and method to achieve consensus between reviewers, inclusion and exclusion criteria, characteristics of the videos reported, ethical oversight, and follow-up.

Results. Thirty-three studies were identified. Most were recent and published in specialty journals. Typically, these included more than 100 videos, and were examined by multiple reviewers. Most studies described characteristics of the videos, number of views, and sometime characteristics of the viewers. Accuracy of portrayal of the health issue under consideration was a common focus.

Conclusion. Optimal transparency and reproducibility of studies of YouTube health-related videos can be achieved by following guidance designed for systematic review reporting, with attention to several elements specific to the video medium. Particularly when seeking to replicate consumer viewing behavior, investigators should consider the method used to select search terms, and use a snowballing rather than a sequential screening approach. Discontinuation protocols for online screening of relevance ranked search results is an area identified for further development.

Introduction

Social media provides effective forums for consumer-to-consumer knowledge exchange and sharing of health information. As well, it is an avenue for health care providers to potentially influence care. An American survey of cancer patients showed that 92% believe that internet information empowers them to make health decisions and helps them to talk to their physicians (McMullan, 2006). Social media is increasingly used by consumers, particularly young adults, (Fox & Jones, 2009) and parents (Moore, 2011).

YouTube is a video-sharing web site that has found a place in health information dissemination. It has been used in medical education (Wang et al., 2013), patient education about specific conditions (Mukewar et al., 2012) and health promotion (O’Mara, 2012). Misinformation has also been shared (Syed-Abdul et al., 2013) and the possibility of covert industry influence has been suggested (Freeman, 2012), leading to guidelines for assessing the quality of such videos (Gabarron et al., 2013).

At present, little is known about the impact of social media and video sharing on pain management practices. The casual searching and viewing of vaccination videos on YouTube revealed a number of “home videos” of infants receiving vaccinations and demonstrated that poor pain management during immunizations is common. We wished to conduct a systematic review of YouTube videos depicting infants receiving immunizations to ascertain what pain management practices parents and health professionals use to reduce immunization pain and distress.

Systematic reviews synthesize research evidence using formal methods designed to safeguard against epidemiological bias. They are reported in a transparent manner that allows the reader to assess the robustness of the study and replicate it. There are various approaches to systematic reviews. These include meta-analyses, in which results are synthesized statistically (Moher et al., 2009), as well as qualitative and mixed methods systematic review (Wong et al., 2013). Kastener argues that “by matching the appropriate design to fit the question, synthesis outputs are more likely to be relevant and be useful for end users (Kastner et al., 2012)”.

It was our intent to adapt this versatile methodology to systematically review YouTube videos of infant vaccination. However, YouTube was expected to pose some particular challenges to systematic inquiry.

Systematic reviews typically synthesize research articles and reports. This evidence base is relatively stable and easily captured and manipulated, with metadata that can be retrieved from bibliographic services such as PubMed or Ovid MEDLINE. In the traditional model of systematic reviews, the body of knowledge is assumed to change, but there is a tacit assumption that the change is through the addition of new evidence. Indeed, little is removed from or modified in the corpus of published scientific literature. As of late April 2013, MEDLINE contained 949,881 with a publication year of 2012, of which only 525 represent retraction notices and 45 represent published erratum.

In contrast, the web and video sharing services such as YouTube are dynamic. Videos can be added or removed at any time by their publishers (or by the host, for violations of copyright or community guidelines), and the order of material in search results may change from day to day. The phenomenon of web resources disappearing is known as “decay” or “modification” (Bar-Ilan & Peritz, 2008; Saberi & Abedi, 2012).

Recognizing that we would not be able to capture and study all YouTube videos ever posted on our topic, we instead sought to craft an approach that would let us capture the cohort of the videos in the YouTube domain on a given day, and extract the relevant information quickly to avoid the loss of any relevant videos.

This paper represents the findings of a preliminary step in designing our systematic review. We surveyed published studies of health-related YouTube videos to address the following question: In reviews and systematic reviews of health-related YouTube videos, what are common methodological challenges, exemplary methods and optimum reporting practices?

Methods and Materials

PubMed was searched April 20, 2012, using the term “YouTube”; the search was limited to the Systematic Review subset. This yielded only 4 records, only 2 of which appeared to be reviews, so the limit was withdrawn. This expanded the search result to 153 records, with the earliest publication occurring in the spring of 2007.

A second approach, a PubMed search of “YouTube and (search or methods)” yielded 86 records. The sample was augmented with two additional reviews nominated by the review team: an early review focusing on the portrayal of vaccinations that we were already aware of, and a review conducted at our institution, which was in press at the time of the search. Just prior to submission of this manuscript, an update search was conducted in PubMed for the term “YouTube” and publications added since the first search were identified and examined for novel features seen infrequently or not at all in the original sample (Table 1).

Table 1 Electronic search strategy.

Main search		
Interface and search date	PubMed, April 2012	
Search string	YouTube and (search or methods)	
Yield	86 records	
Update search		
Interface and search date	PubMed, November 22, 2012	
Search string	YouTube	
Yield	Records were screened by date, newest to oldest, until reaching the newest article included from the original search
(Pant, added to PubMed 2012/04/11).
Yield: 46 records	

The search results were screened by a single reviewer using the following criteria:

The videos reviewed focused on consumer health rather than targeted toward health care providers or trainees and the video did not focus on adoption or use of social media. No limits were imposed regarding publication date or language. Data extracted from these studies included type of journal (general medical, specialty medical journal or internet/social media journal), topic of the review, characteristics of the search, methods of review including number of reviewers and method to achieve consensus between reviewers, inclusion and exclusion criteria, characteristics of the videos reported, ethical oversight, and follow-up.

Data were extracted from the published report – we did not contact authors to seek clarification of methods. As we focused on methodological aspects as reported, we did not perform additional risk of bias assessments on the individual studies, did not plan to perform meta-analysis, and did not publish a protocol.

Results and Discussion

Twelve eligible studies were identified from the initial search (Fig. 1) (Backinger et al., 2011; Pant et al., 2012; Ache & Wallace, 2008; Tian, 2010; Lo, Esser & Gordon, 2010; Steinberg et al., 2010; Knösel, Jung & Bleckmann, 2011; Singh, Singh & Singh, 2012; Pandey et al., 2010; Knösel & Jung, 2011; Keelan et al., 2012; Fat et al., 2012). Topics of the 12 initial reviews were: smoking cessation (Backinger et al., 2011), acute myocardial infarction (Pant et al., 2012), HPV vaccination (Ache & Wallace, 2008), organ donation (Tian, 2010), epilepsy (Lo, Esser & Gordon, 2010), prostate cancer (Steinberg et al., 2010), dentistry (Knösel, Jung & Bleckmann, 2011), rheumatoid arthritis (Singh, Singh & Singh, 2012), H1N1 (Pandey et al., 2010), orthodontics (Knösel & Jung, 2011), vaccination (Keelan et al., 2012), and Tourette syndrome (Fat et al., 2012). Most (8) were published in specialty journals (Lo, Esser & Gordon, 2010; Steinberg et al., 2010; Knösel, Jung & Bleckmann, 2011; Singh, Singh & Singh, 2012; Knösel & Jung, 2011; Fat et al., 2012; Pant et al., 2012; Backinger et al., 2011). Three were in general and internal medicine journals (Ache & Wallace, 2008; Keelan et al., 2012; Pandey et al., 2010), one in a health communications journal (Tian, 2010). Most were published from 2010 to 2012. The earliest was 2008 – three years after YouTube’s inception (Ache & Wallace, 2008). Results are summarized in Table 2.

Figure 1 PRISMA flow diagram for included studies.

Adapted from: Moher D, Liberati A, Tetzlaff J, Altman DG, The PRISMA Group (2009). Preferred Reporting Items for Systematic Reviews and Meta-Analyses: The PRISMA Statement. PLoS Med 6(6): e1000097. DOI 10.1371/journal.pmed1000097.

Table 2 Characteristics of 12 studies of YouTube consumer health videos with PRISMA.

Characteristic	N
Total = 12	%	
Type of journal (G = general/internal medicine,
S = specialty, I = internet/social media journal)	S = 8, G = 3, I = 1	
Year of publication: median (range)	2011 (2008–2012)	
Searcha			
Search date given	10	83	
Number of terms searched: median (range)	3 (1–5)		
Direct search of YouTube	12	100	
Source of terms explained	1	8	
Used multiple searches or samples	3	25	
Videosb			
Number of videos included	Mean 145	
	Median 112	
Inclusion criteriac			
English only	8	67	
“Off topic” excluded	9	75	
Descriptive characteristics collectedd			
Number of views	12	100	
Length	8	67	
Date posted	5	42	
Number of “Likes”	3	25	
Average rating score	3	25	
Number rated by viewers	2	17	
Intended audience	2	17	
Production quality (Amateur/Pro)	2	17	
Review methode			
Qualifications of reviewer described	6	50	
2 or more reviewers	10	83	
Resolution method described	6	50	
Kappa reported	7	58	
Training of reviewers described	2	17	
Blinding of reviewers	2	17	
Notes.

The reader wishing guidance on these aspects of reporting may wish to consult Preferred reporting items for systematic reviews and meta-analyses: the PRISMA statement (Moher et al., 2009) and the accompanying elaboration and explanation (Liberati et al., 2009).

a PRISMA element 7 and 8.

b PRISMA element 17.

c PRISMA element 6.

d PRIMSA element 11.

e PRISMA elements 9 and 10.

Thirteen additional reviews were identified from the update search (Fig. 2). Three were found ineligible when the full text of the article was examined, ten were found eligible (Kerson, 2012; Richardson & Vallone, 2012; Stephen & Cumming, 2012; Jurgens, Anderson & Moore, 2012; Thomas, Mackay & Salsbury, 2012; Ehrlich, Richard & Woodward, 2012; Tourinho et al., 2012; Mukewar et al., 2012; Kerber et al., 2012; Clerici et al., 2012) and an eleventh (Bromberg, Augustson & Backinger, 2012), cited by one of the ten as informing their methods (Richardson & Vallone, 2012).

Figure 2 PRISMA flow diagram for studies from the updating seach (Supplemental Information 1).

Adapted from: Moher D, Liberati A, Tetzlaff J, Altman DG, The PRISMA Group (2009). Preferred Reporting Items for Systematic Reviews and Meta-Analyses: The PRISMA Statement. PLoS Med 6(6): e1000097. DOI 10.1371/journal.pmed1000097.

The searches

All reviews searched directly on the YouTube site, rather than through a third party interface such as Google advanced search. Ten of the 12 reviews reported the date of the search (Ache & Wallace, 2008; Pandey et al., 2010; Fat et al., 2012; Backinger et al., 2011; Keelan et al., 2012; Pant et al., 2012; Steinberg et al., 2010; Knösel & Jung, 2011). Most included several terms in the search, and these were presumably linked with “OR”. Most did not address the sort order, so presumably used the default values. Currently, YouTube search results are sorted by relevance as the default. One review sampled the top ranked items from searches sorted by relevance and number of views, using the default of searching all of YouTube, and then again searching only those classified by the person who posted the video as “educational” (4 samples in all) (Knösel, Jung & Bleckmann, 2011). Only one review explained how search terms were selected – by using Google Trends to determine which topical terms were most searched (Backinger et al., 2011). In the updated set, two additional studies used empirically derived search terms – most common brands and common search terms from Google Insights (Richardson & Vallone, 2012; Bromberg, Augustson & Backinger, 2012).

Several reviews attempted to make the search realistic, that is, searching as consumers might search (Fat et al., 2012; Pant et al., 2012; Knösel & Jung, 2011; Backinger et al., 2011); but all seemed to have worked from the search list rather than using a snowball technique (Grant, 2004). Snowballing is a technique used in sampling for qualitative studies - cases with connection to other cases are identified and selected (Giacomini & Cook, 2000) and in information retrieval - references to references are considered for relevance (Greenhalgh & Peacock, 2005). It is a useful adjunct when identifying all relevant candidates through a search engine is difficult for whatever reason (Horsley, Dingwall & Sampson, 2011).

However, three of the searches from the updated search did describe snowballing, as follows: “As clips were viewed, additional suggestions were offered by the site and these in turn led to further suggestions” (Stephen & Cumming, 2012). “For each of the top 10 videos, the top three related videos (ranked by YouTube) were also coded” (Thomas, Mackay & Salsbury, 2012). Finally “The search was supplemented by also reviewing the list of featured videos that accompany search results” (Kerber et al., 2012).

Only one review imposed filters on the search (in that case, that the video had been uploaded in the past three months) (Pandey et al., 2010).

The inclusion criteria

Eight of the 12 reviews stated that only English language videos were included (Backinger et al., 2011; Keelan et al., 2012; Singh, Singh & Singh, 2012; Lo, Esser & Gordon, 2010; Pant et al., 2012; Steinberg et al., 2010; Pandey et al., 2010; Tian, 2010). None of the other 4 reported that they were language inclusive. Nine reported that they excluded “off topic” videos (Keelan et al., 2012; Singh, Singh & Singh, 2012; Pant et al., 2012; Steinberg et al., 2010; Pandey et al., 2010; Tian, 2010; Fat et al., 2012; Knösel, Jung & Bleckmann, 2011; Ache & Wallace, 2008), but few gave clear criteria defining what was “on topic”. Eight reported that they removed duplicates (Singh, Singh & Singh, 2012; Pant et al., 2012; Steinberg et al., 2010; Pandey et al., 2010; Tian, 2010; Fat et al., 2012; Ache & Wallace, 2008; Backinger et al., 2011). Details of the treatment of duplicates were sparse; for instance, none stated how they selected which version to keep, or whether they aggregated view counts across all versions - although two stated that if videos had multiple parts, only one was kept, and both of these stated that they averaged views across all parts (Singh, Singh & Singh, 2012; Pant et al., 2012). Two reviews stated that the video must have sound to be eligible (Pant et al., 2012; Steinberg et al., 2010). One included only videos under 10 min in length (Steinberg et al., 2010) and one excluded videos blocked by their institutions’ internet filters (Backinger et al., 2011).

Reported characteristics of the review

Descriptive characteristics: All reviews reported on some characteristics of the videos. Elements most commonly reported were: number of views (Pant et al., 2012; Steinberg et al., 2010; Pandey et al., 2010; Keelan et al., 2012; Ache & Wallace, 2008; Backinger et al., 2011; Lo, Esser & Gordon, 2010; Knösel, Jung & Bleckmann, 2011; Fat et al., 2012; Singh, Singh & Singh, 2012; Tian, 2010; Knösel & Jung, 2011), length in minutes (Pant et al., 2012; Lo, Esser & Gordon, 2010; Steinberg et al., 2010; Knösel, Jung & Bleckmann, 2011; Singh, Singh & Singh, 2012; Pandey et al., 2010; Knösel & Jung, 2011; Keelan et al., 2012), and date posted (Pant et al., 2012; Lo, Esser & Gordon, 2010; Tian, 2010; Singh, Singh & Singh, 2012; Pandey et al., 2010). While most reported median, or mean number of views, often with some measure of dispersion, one reported the concentration of views – five videos accounted for 85% of total views (Kerber et al., 2012).

Other characteristics reported included number of “likes” (Pant et al., 2012; Singh, Singh & Singh, 2012; Fat et al., 2012), rating score (Steinberg et al., 2010; Tian, 2010; Keelan et al., 2012), times rated by viewers (Lo, Esser & Gordon, 2010; Tian, 2010), intended audience (Pant et al., 2012; Steinberg et al., 2010), amateur/pro, based on production quality (Fat et al., 2012; Lo, Esser & Gordon, 2010), type if non-standard (i.e., song, animation, advertisement) (Pant et al., 2012), country of origin or address of author (Tian, 2010). Importantly, one from the original set (Fat et al., 2012) and three from the update set, harvested self-reported demographics of viewers (Mukewar et al., 2012; Stephen & Cumming, 2012; Richardson & Vallone, 2012).

Several classified the videos according to the creating source (Pant et al., 2012; Ache & Wallace, 2008; Singh, Singh & Singh, 2012; Pandey et al., 2010; Knösel & Jung, 2011), each used its own typology, but common elements were: personal experience/patient, news reports, professional associations, NGOs such as WHO or Red Cross, pharmaceutical companies and medical institutions. Three reviews from the update set addressed the issue of covert advertising – two for a tobacco product (Richardson & Vallone, 2012; Bromberg, Augustson & Backinger, 2012), the other discussed the notion of paid testimonials appearing as consumer-posted videos (Mukewar et al., 2012).

Sample size

The number of videos assessed ranged from 10 to 622, with a mean of 145 and median 112. Some screened the entire search results (maximum 1634 videos). More common was an approach of taking a fixed sample size and screening this set, retaining those eligible after duplicates, off topic and other ineligible material was removed. Two reviews used a fixed sample size (Knösel, Jung & Bleckmann, 2011; Lo, Esser & Gordon, 2010). Several set a fixed sample size to screen (Fat et al., 2012; Knösel & Jung, 2011; Backinger et al., 2011). No reviews reported a formal sample size calculation.

Review methods

Two reviews reported saving all eligible videos offline (Ache & Wallace, 2008; Pandey et al., 2010). Some reported viewing, screening or assessing online, at the time of discovery. Two (both by Knosel) described the reviewing conditions in some detail; videos were viewed at the same time and under the same conditions by two assessors (Knösel, Jung & Bleckmann, 2011; Knösel & Jung, 2011). Knosel also described opportunities for the reviewers to communicate – required in one review (Knösel, Jung & Bleckmann, 2011) and prevented in the other (Knösel & Jung, 2011).

Eight of the 12 reviews described the reviewers (Backinger et al., 2011; Lo, Esser & Gordon, 2010; Keelan et al., 2012; Steinberg et al., 2010; Knösel & Jung, 2011; Singh, Singh & Singh, 2012; Knösel, Jung & Bleckmann, 2011; Fat et al., 2012). Most were health care professionals, however, one used lay raters – one potential patient (a youth) and one parent – to gain their perspective (Knösel & Jung, 2011).

Ten of 12 reported on the number of reviewers – 8 reported using 2 reviewers for each video (Backinger et al., 2011; Tian, 2010; Singh, Singh & Singh, 2012; Knösel, Jung & Bleckmann, 2011; Fat et al., 2012; Pandey et al., 2010; Keelan et al., 2012; Steinberg et al., 2010), one reported 3 (professional, parent, youth) (Knösel & Jung, 2011) and one implied multiple reviewers without specifying the number (Ache & Wallace, 2008). No review reported having only one reviewer make assessment of content. Four of the 10 with multiple reviewers reported using a third reviewer as arbitrator (Keelan et al., 2012; Singh, Singh & Singh, 2012; Steinberg et al., 2010; Backinger et al., 2011). Seven of the 10 computed kappa on reviewer agreement. It was not always made clear which rating was used if conflicts occurred – i.e., neither arbitration nor consensus was described (Fat et al., 2012; Keelan et al., 2012; Pandey et al., 2010; Knösel, Jung & Bleckmann, 2011; Steinberg et al., 2010; Tian, 2010; Backinger et al., 2011).

Only two reviews described a training or calibration exercise prior to undertaking assessments. Backinger described 4 h of training in which definitions were discussed, and 5 practice videos coded (Backinger et al., 2011). Tian described pre-testing their code book, using 20 videos and 40 text comments (Tian, 2010).

Blinding was used in two reviews. Pandey reported that reviewers were blind to the purpose of the study (Pandey et al., 2010). Lim Fat reported, “The individual who rated the comments was blinded to the classification of the video as being a positive, negative or neutral portrayal of Tourette syndrome. Likewise, the raters for classification of the videos were blinded to the classification of the comments” (Fat et al., 2012).

Ethical oversight

None of the original cohort of twelve reviews stated that IRB approval was obtained. One review explicitly stated that IRB approval was deemed unnecessary due to the nature of the study (Fat et al., 2012). In the 11 additional studies reviewed, four made explicit statements about IRB approval, one sought approval (Ehrlich, Richard & Woodward, 2012), three stated they were exempt (Richardson & Vallone, 2012; Mukewar et al., 2012; Kerber et al., 2012).

Follow-up

Three videos examined a cohort of reviews at two or more points in time looking changes in the number of hits. One review from the original set reported a follow-up at one and six months (Lo, Esser & Gordon, 2010). Two more from the update set included follow-up, one after 7 months (Mukewar et al., 2012) and one after 1 month and 7 months (Stephen & Cumming, 2012).

Outcomes of interest

The characteristics of the videos that were evaluated varied depending on the study objectives, but videos were commonly assessed as providing true or reliable information or being positive or negative toward the health issue addressed. One review from the original set (Singh, Singh & Singh, 2012) used a validated scale as part of the assessment - DISCERN: an instrument for judging the quality of written consumer health information on treatment choices (Charnock et al., 1999). One review identified from the update search (Mukewar et al., 2012) used two scales adapted from Inflammatory Bowel Disease patient education web sites; a detailed scale specific to IBD, and a 5-point global quality score.

Two looked at knowledge translation – one of these examined whether videos posted after a change in guidelines for cardiopulmonary resuscitation employed the new or old standard (Tourinho et al., 2012). Another investigated the integrity with which parents and carers implement the Picture Exchange Communication System (PECS), in a naturalistic setting (Jurgens, Anderson & Moore, 2012).

One review assessed the findability of the videos – having identified 33 videos that portrayed complete Epley maneuvers for benign paroxysmal positional vertigo, they searched again using very general terms – dizzy, dizziness, vertigo, positional dizziness, positional vertigo, dizziness treatment, and vertigo treatment. The investigators then determined where or if one of the videos depicting the Epley maneuver appeared in the list of relevance-ranked results for the less specific terms (Kerber et al., 2012).

Discussion

These reviews are recent, and for the most part, clearly reported. There are examples of excellent reporting in most facets of the review – study question, inclusion and exclusion criteria, search strategy, screening and data extraction methods. Few reviews, however, reported all elements well. Improved reporting would increase transparency and allow the reader to better assess the risk of bias in the study design (Tricco, Tetzlaff & Moher, 2010). The study design should be strong and reproducible, with methods in line with those of a well-conducted systematic review (Moher et al., 2009). Through this manuscript, we aim to describe the array of methods and data available to those planning to undertake this type of work, recognizing that, depending on the data examined and the objectives of the review, methods will vary.

While it is premature to put forth reporting guidelines – defined as a minimum set of elements that should be reported to enable the reader to understand the conduct of the study, assess the risk of bias and generalizability of results, the PRISMA checklist (Moher et al., 2009) and accompanying elaboration and explanation (Liberati et al., 2009) generalizes to many aspects of video reviews. The elements that have no real parallel in systematic reviews of research studies warrant the most consideration and some of these are elaborated in Table 3.

Table 3 Some systematic review methodological considerations specific to review of consumer health videos, with examples.

Characteristic	Examples	
Whether the search was intended to identify all consumer-oriented videos or a sample	We reviewed videos posting: on YouTube; on the web.	
What video sources were selected	YouTube; Vimeo; Yahoo Video	
How search terms were derived	Search terms were chosen; by the investigator; by soliciting suggestions from consumers; based by search log data such as Google Trends	
Any system preferences that would have influenced the search results	What sort order was used; the search was limited to reviews classification as “educational”; the search was limited to recently added videos	
How the review of the search results was conducted	Sequential screening of search results; snowballing	
Discontinuation rules	Results were screened: until a predetermined sample size was obtained (state how the sample size was determined); until the entire search result was considered; until predetermined discontinuation criteria were met (state how that critera was determined).	
How the instability of rankings was addressed	All screening done in a single day; Search results were captured for later assessment.	
Any other measures designed to neutralize bias in the identification of videosa	We using a computer outside the institutional firewall and not previously used to search YouTube; We searched through DuckDuckGo.com to avoid having our location influence the ranking of videos.	
Notes.

a Many search sites customize search results based on factors such as your geographic location and search history (Pariser, 2013).

A number of factors make video searches less stable, and thus less replicable, than the sorts of database searches used in systematic reviews where results either match the search criteria or do not and results are sorted by date. In any relevance-ranked search results, the order will change as new entries are added, existing ones removed, or the proprietary ranking algorithm is changed. Thus, a systematic approach that will accomplish the goals of the review (whether they are exhaustive identification of eligible videos or a sample of fixed size that represents the videos that the target audience is most likely to find) is needed, and should be fully described.

There was one aspect of our upcoming review of immunization videos that was not informed by this survey of published studies - a discontinuation rule to stop screening when few additional studies were being found. Our initial search yielded 6,000 videos. Unlike the searches of bibliographic databases used for study identification in systematic reviews, this search result was ranked according to relevance. Spot checks showed that most of the lower ranked videos were irrelevant. Given the size of the list, screening the entire list in one sitting was not feasible. We did not know of a way to “download” the list, and had no assurance that the list would remain unchanged on subsequent screening days. Thus, a protocol was needed to discontinue screening when further screening was unlikely to yield additional eligible videos.

Discontinuation rules would allow one to manage relevance-ranked search results, and are essential when screening web search results that are often large, cannot be easily captured as a whole, and are not static from day to day. Given the absence of empirical guidance, we devised a pragmatic rule: screen until twenty consecutive ineligible videos are reviewed, then assess a margin of 50 more. Depending on the number of eligible videos found in the margin, a decision could be taken to stop screening or continue. With this discontinuation rule, one needs to accept that some additional eligible videos might have been found had the entire retrieval been screened, however, the likelihood of missing a large number is low given the relevance ranking.

Examples of discontinuation rules can be found in several health care fields. Computerized adaptive testing often use validated stopping rules to discontinue the test when it becomes statistically unlikely that administering additional items would improve the accuracy of the assessment (Babcock & Weiss, 2009). Clinical trials that have planned interim analyses may have pre-specified stopping rules designed “to ensure trials yield interpretable results while preserving the safety of study participants” (Cannistra, 2004). As search engine ranking algorithms improve, there is increasing opportunity for systematic reviewers to use sources that offer relevance ranking, such as Google Scholar or PubMed Related Citations. Experimental research has demonstrated that ranking algorithms can successfully place eligible records high in a search result (Cohen et al., 2006; Sampson et al., 2006). Yet, we were unable to identify any practical guidance on stopping rules for screening in the systematic review context, nor any explicit reports of when or if screening was stopped for Internet-based searches in systematic reviews. This is an area requiring more complete reporting on the part of systematic reviewers, as well as a useful area for further research.

It should be noted that factors such as screening order, the use of snowballing or other techniques to mimic consumer searching behaviour, and discontinuation rules are relevant only when there is a tacit acceptance that not all potentially relevant videos will be identified.

While we have focused our efforts on informing the conduct of systematic reviews, video producers hoping to reach consumers may wish to consider several factors. As the difference in number of views varies 1000 fold, a clear marketing plan is needed for any production effort to be worthwhile. Our review suggests that videos styled as home videos appeal to a broader audience than dyadic videos. As much as reviewers use empirically defined search terms, producers will want to select titles and keywords that are likely to match what consumers type in to the search bar. Producers will need to consider the factors that rank a video high in the related list as well as those that will make it appear in the search results. As the ranking algorithms for both search engine ranking and related sidebar ranking are proprietary and subject to change, video producers will want to seek up-to-date guidance on “Search Engine Optimization” for YouTube, or any other video channel they intend to use.

Limitations of this systematic review include the fact that there may be additional informative reviews that we did not identify and include. We only searched one traditional bibliographic database, and did not include social media such as blog postings. Also, reviews of consumer health videos are relatively new. YouTube was created in 2005, and given the time needed to conduct and publish reviews; there may be a large number in preparation. Certainly, as many appeared in the course of this project (April to November 2012) as were published prior to its start, suggesting that there may be innovations that we have not yet captured.

Conclusions

There are many gaps in reporting in these early studies of YouTube videos, and no known reporting standards. Some strong trends are apparent – reviewers use simple searches at the YouTube site, with restriction to English with some process to remove off-topic retrievals, duplicates and select only one of multi-part videos. Two reviewers are generally used and kappa is commonly recorded. Although reviewers often state they are attempting to mimic user behaviour, this is generally limited to including the first few pages of search results – only a few of the most recent reviews have used a more sophisticated snowballing approach. Selection of search terms is typically done by health care professionals, whose searches may be quite different from the searches that consumers would typically do; however, there are examples of empirically determined search terms. As well, health consumers are infrequently included as assessors. Finally, efficiencies can be gained by determining stopping criteria for screening large relevance ranked search results such as those provided by YouTube. In the absence of formal reporting guidelines (which might be premature), we recommend that those wishing to review consumer health videos use accepted systematic review methods as a starting point, with some of the elements specific to the video medium that we describe here.

Supplemental Information

Supplemental Information 1 PRISMA checklist

Click here for additional data file.

Additional Information and Declarations

Competing Interests

Author Contributions

Data Deposition

Margaret Sampson is an Academic Editor for PeerJ.

Margaret Sampson conceived and designed the experiments, performed the experiments, analyzed the data, wrote the paper.

Jordi Cumber conceived and designed the experiments, performed the experiments, wrote the paper.

Claudia Li, Catherine M. Pound, Ann Fuller and Denise Harrison conceived and designed the experiments, wrote the paper.

The following information was supplied regarding the deposition of related data:

Dryad: http://doi.org/10.5061/dryad.4jh42.

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
