# Peer review of "A systematic review of methods for studying consumer health YouTube videos, with implications for systematic reviews"

_PeerJ, doi:10.7717/peerj.147_

## Round 0.1 · original submission · Minor Revisions

As with the two reviewers, I believe this article has merit and should be accepted with revisions. Understanding how to evaluate consumer health videos is an important area of study which will only grow . Please be sure to copy edit the manuscript as there are several errors. I have three broad based suggestions (some corresponding to reviewers) that may improve the manuscript for the reader and several smaller ones. First, the introduction appears to focus more of the background of how you came to the decision to study health videos by focusing on infant pain which is never mentioned again. It would be helpful to discuss in more detail about who is creating these videos, more on why this is an important area of study and most importantly to bring it back to your conclusion and goals – what are the current accepted systematic review methods (lines 316) and why we need to modify these current methods for videos and create a new method. In the method section, please merge how you searched into one paragraph (e.g. add 61-63 into the paragraph). Please define “snowball techniques” in more detail. Line 138 appears to be missing some text. The review provided so much important information as presented in the table and was a well done descriptive study. As with reviewer two, it would be nice to see you expand the discussion and conclusions based on your knowledge and findings to include – what are the most important areas that researchers cover when conducting reviews of Web-based health videos beyond the descriptive table of what you covered, why it is important and include a sample rubric, and with your knowledge, it would be very interesting to include a brief paragraph on what individuals developing videos should take into account - though this is clearly a separate paper and just a suggestion to enhance the paper for a larger audience. Looking forward to seeing the revised manuscript.

Reviewer 1 ·

Basic reporting

While I am unfamiliar with the journal, the research question seems well-identified, contained, and sized.

Experimental design

The research question is clear. This study is strictly descriptive, which is fine for addressing the authors' research question. Methods were well-described.

Validity of the findings

It seems likely that these findings are valid, inasmuch as the methods for the present review are well-described.

Additional comments

This paper is a useful description of “reviews of reviews” of YouTube videos related to health. It specifies many dimensions of review reporting, and in some cases makes recommendations (e.g., for stopping rules). As information sources grow in channel types, attempts to summarize it will need to develop standards for reporting, and papers such as this one will help to make this happen.
It is interesting that some institutions required IRB review for the studies and some did not; were there differences between the two categories that account for this?
One issue that puzzles me a bit has to do with “snowballing” to include videos that appear as recommendations alongside the “index” video. It seems that using this procedure would result in non-replicable results (if replicability is even a relevant issue here), and would interfere with systematicity in the search. A description of how this strategy could be made more systematic would be helpful.
I would request that the authors attend carefully to their use of the terms “review” and “study”. Sometimes reviews are called studies, and this is confusing: for example, in line 260 where the study that is referred to is the pain-during-vaccination one that initiated this project.

Reviewer 2 ·

Basic reporting

No comment

Experimental design

No comment

Validity of the findings

No comment

Additional comments

This article addresses an important topic. Although the authors write in their conclusion that formal reporting guidelines might be premature, it might be helpful to provide a flexible sample rubric or an adaptable list of elements to include and/or report. This sample list could be designed in the spirit of the Dublin Core which enables archivists to classify items according to a flexible list of accepted search terms, which can be adapted to fit the specifics of various projects by content and context.

---

## Round 0.2 · Minor Revisions

Thank you for making the edits to the document. It enhances the manuscript. My main concern at this point is the introduction. It is now extremely short and the paragraphs do not necessarily flow after items were deleted without modifying the text - and new text was not included. I apologize if this was not clear in the previous edit but is important for the best possible manuscript. I believe the introduction should at minimum:

1- have an additional paragraph on how health videos are being used presently across health disciplines and the information/intervention they are attempting to convey to consumers. For example, it is post-op care instructions, general health advice, etc. Please give examples.Even though this is a description of reviews this will help orient the reader who may not be experts.

2- Your main question is the following: "In reviews and systematic reviews of health-related YouTube videos, what are common reporting practices, exemplary methods and methodological challenges."

After the above inserted paragraph, please write an additional paragraph on the problem or challenge you are trying to address with your paper -- what discrepancies exist, why conduct this review aside from the prevalence of these videos for example. More information on the importance of this review is also important to orient the reader.

In general the background should be clearer and set the reader up for your methods.

Based on all your research, this should be a minor revision. Please contact me with questions. Fred

---

## Round 0.3 · accepted · Accept

Thank you for making the changes to the introduction - the paper will be useful to help understand this research area. The paper can be accepted but please make the following changes:
1. Please copy edit the entire manuscript. There were several typos
2. Line 24: At present little is known about.... I would start the paragraph with - line 27 - We wished to conduct a systematic review -- first. Then At present... (Switch order). That will help transition the reader - then leading to why you did this preliminary paper.

Congratulations.